# The Pork Meat or the Environment of the Production Facility? The Effect of Individual Technological Steps on the Bacterial Contamination in Cooked Hams

**DOI:** 10.3390/microorganisms10061106

**Published:** 2022-05-27

**Authors:** Helena Veselá, Kateřina Dorotíková, Marta Dušková, Petra Furmančíková, Ondrej Šedo, Josef Kameník

**Affiliations:** 1Department of Animal Origin Food and Gastronomic Sciences, Faculty of Veterinary Hygiene and Ecology, University of Veterinary Sciences Brno, 612 42 Brno, Czech Republic; veselah@vfu.cz (H.V.); dorotikovak@vfu.cz (K.D.); furmancikovap@vfu.cz (P.F.); kamenikj@vfu.cz (J.K.); 2Central European Institute of Technology, Masaryk University, 625 00 Brno, Czech Republic; ondrej.sedo@ceitec.muni.cz

**Keywords:** fresh meat, tumbled meat, microbiota, lactic acid bacteria, MALDI-TOF MS, incubation temperature

## Abstract

The aim of this study was to analyse the influence of the contamination level of fresh meat on the bacterial population in raw material before cooking and on the microbiota of cooked hams following heat treatment. The effect of incubation temperatures of 6.5 and 15 °C on the results obtained was also evaluated during the bacteriological investigation. The total viable count (TVC), the number of *Enterobacteriaceae* and lactic acid bacteria (LAB) were determined in the samples. LAB were isolated from 13 samples out of the 50 fresh meat samples. The species most frequently detected included *Latilactobacillus sakei*, *Leuconostoc carnosum*, *Enterococcus gilvus*, *Latilactobacillus curvatus*, and *Leuconostoc gelidum*. The meat sampled after the brine injection and tumbler massaging showed higher bacterial counts compared to fresh meat samples (*p* < 0.001). The heat treatment destroyed the majority of the bacteria, as the bacterial counts were beneath the limit of detection with a few exceptions. Although the primary cultivation of samples of cooked hams did not reveal the presence of LAB, their presence was confirmed in 11 out of 12 samples by a stability test. Bacteria of the genus *Leuconostoc* were the most numerous.

## 1. Introduction

Heat treatment is a critical technological step in the production of cooked hams. When the temperature at the core of the products reaches 65–75 °C, most of the vegetative bacteria are killed [1,2]. Therefore, heat treatment is usually defined as a critical control point in HACCP plans with a fundamental effect on the safety of final products. The recommended time/temperature combination of 70 °C for 2 min reduces the number of the most heat resistant vegetative bacterium *Listeria monocytogenes,* by more than 6 log orders [3]. Nevertheless, pronounced growth of bacteria occurs in cooked hams towards the end of their shelf life regardless of the combination of hygiene measures and the use of measures to extend shelf life such as refrigeration, microaerophilic conditions, and the presence of salt and nitrite [1].

There are several groups of organisms that potentially contribute to spoilage of meat products under appropriate conditions. The main bacterial group associated with the spoilage of cooked meat products, including cooked hams, are lactic acid bacteria (LAB) [4,5,6]. Their growth is favoured by combinations of microaerophilic conditions in the product, by the presence of sodium chloride and sodium nitrite and by a reduced water activity [7]. However, it is yet not fully understood whether bacteria that are present in meat products originate from the meat itself or rather from handling-related operations in the processing line or from the environment [8].

Cooked hams are sold in retail as modified atmosphere packaged (MAP) slices and the present microbiota has a fundamental effect on their shelf life. Since only a few bacteria survive the heat treatment, the actual microbial contamination occurs in post-cooking operations, most importantly the packaging process [8]. Bacteria get onto the product during the removal of the casing before slicing, during slicing itself and during subsequent packaging. The hygiene of the plant, including maintenance of personal hygiene principles in food handlers, is of crucial importance at this stage [9]. Although secondary contamination of the surface of cooked hams during packaging may be important to their subsequent shelf life, microbiota that has survived heat treatment may influence the shelf life more than expected. Kameník et al. [9] investigated 6 samples of whole cooked hams intended for slicing. A bacterial population excessing 5 log CFU/g (colony-forming units) was demonstrated in one sample. The shelf life of MAP sliced cooked hams is generally 21–28 d, though it may be as long as 45 d [1]. Producers endeavour to offer products with the longest possible shelf life as it gives them a competitive advantage on the market.

Also, incubation temperature is an important parameter influencing the results of bacterial analysis of either raw meat or cooked hams. Dušková et al. [10] reported a better recovery for LAB incubated at a temperature of 15 °C (favouring the growth of psychrotrophic bacteria) compared to a temperature of 30 °C, which is the optimal growth temperature for mesophiles. Similarly, Pothakos et al. [11] pointed to the fact that a cultivation temperature of 30 °C does not provide an entirely objective picture of the microbiota present, when evaluating the level of spoilage bacteria in products stored under refrigerated conditions. Thus, in order to obtain adequate representation of LAB occurring in meat products, it is crucial to determine proper cultivation temperature as well.

The aim of this study was to analyse the influence of the level of contamination of fresh meat on the bacterial population in raw production material before heat treatment and on the microbiota of cooked hams following heat treatment. The effect of incubation temperatures of 6.5 and 15 °C on the results obtained was also evaluated during the bacteriological investigation.

## 2. Material and Methods

### 2.1. Sample Collection

Five collections of samples, consisting of fresh meat (pork topsides) samples, tumbled meat samples and cooked hams samples, took place in the facility of one industrial meat processor during the period March–May 2021. On day 1, the pork topsides samples were obtained. The following day (Day 2), samples of the identical batch of meat after brine injection and tumbling were taken. Samples of the final cooked hams prepared from the meat sampled on the preceding days were taken after another 3 d (Day 5). In total, 10 samples of fresh meat were obtained using a destructive method (sterile forceps, a sterile knife and a sterile template) from an area of 10 cm^2^, twice from each topside sample (pooled sample), on one sampling day. Samples of tumbled meat and hams of a weight of 10 g were taken in a sterile manner from each sample. A total of 50 fresh meat samples, 50 tumbled meat samples, and 30 final cooked hams samples were analysed (Figure 1). All of the samples were transported under refrigeration temperatures to the laboratory on the day of collection. Subsequently, 12 ham samples (from third and sixth sample batch) were further subjected to a so-called “stability test”. The samples were vacuum packed and incubated at 15 °C for 7 d before microbial analysis.

### 2.2. Bacterial Examination

The total viable count (TVC), the number of bacteria of the family *Enterobacteriaceae* and LAB were determined in the samples. A medium with glucose, tryptone, and yeast extract was used for the determination of the TVC (GTY; OXOID, Hampshire, UK). *Enterobacteriaceae* were enumerated on Violet Red Bile Glucose agar (VRBG; OXOID). From each plate, up to 5 colonies were randomly selected for confirmation using oxidase and glucose fermentation tests. LAB cultivation was performed under anaerobic conditions on deMan, Rogosa, and Sharpe agar culture medium (MRS; OXOID). All colonies showing various morphological characteristics were selected from each sample and tested for the presence of catalase and oxidase (JK Trading, Prague, Czech Republic). Due to the nature of the samples, culture media were incubated at 6.5 °C/12 d (TVC and LAB) and 6 d (*Enterobacteriaceae*) and at 15 °C/6 d (TVC and LAB) and 3 d (*Enterobacteriaceae*). Culture media of vacuum-packed ham samples additionally incubated at 15 °C (stability test) were cultivated only at 15 °C.

### 2.3. Bacterial Identification by MALDI-TOF Mass Spectrometry

Isolates with negative oxidase and catalase tests were further identified using matrix-assisted laser desorption ionization – time of flight mass spectrometry (MALDI-TOF MS) following the procedure described by Dušková et al. [12]. The samples for MALDI-TOF MS analysis were prepared by protein extraction (ethanol/formic acid) according to a standard protocol [13]. Mass spectrometry measurements were performed using an Ultraflextreme instrument (Bruker Daltonik, Bremen, Germany) operated in the linear positive ion mode using FlexControl 3.4 software. Mass spectra were processed using BioTyper software (version 3.0; Bruker Daltonik). The identification results were expressed by BioTyper log(scores) indicating the similarity of the unknown MALDI-TOF MS profile to Biotyper database entries (version 10.0; 9607 entries). A BioTyper log(score) exceeding 2.0 indicates a highly confident identification at the species level. A BioTyper log(score) between 1.7 and 2.0 means identification at the species level with lower confidence. Only isolates with a log(score) over 1.7 were taken into account.

## 3. Results and Discussion

### 3.1. Fresh Meat

The results of the bacteriological analyses of meat samples and cooked hams are given in Table 1 and Appendix A. The average contamination of fresh meat was around 3 log CFU in 1 g, which demonstrates a good quality raw material from the microbiological perspective [14]. The company of the samples origin imports pork legs either in the form of whole legs with bone and skin or as boned and skinned legs (4D pork leg). Portioning into individual parts is then performed in the company’s own cutting plant. The sampling conducted in our study included meat that came both from pork legs supplied with the bone and skin and from already boned legs. The results indicate that the meat from legs, which were cut entirely at the company, had a bacterial load approximately 1 log CFU/g lower than the meat from 4D pork legs (data not shown). The results for the number of indicator bacteria from the family *Enterobacteriaceae* and LAB point out good microbiological quality of the fresh meat as well. LAB were isolated from 13 samples out of the 50 samples. The species most frequently detected included *Latilactobacillus sakei* (formerly *Lactobacillus sakei* [15]) isolated from 8 meat samples, *Leuconostoc carnosum* (2 samples), *Enterococcus gilvus* (2 samples), and *Latilactobacillus curvatus* (formerly *Lactobacillus curvatus*) and *Leuconostoc gelidum* (1 sample each).

### 3.2. Tumbled Meat

The meat sampled after the brine injection and tumbler massaging showed higher bacterial counts compared to fresh meat samples, with the differences being statistically significant (*p* < 0.001). While in the case of TVC and *Enterobacteriaceae* there was a numerical increase of 1 log order, in LAB there was an average increase of 2–3 logs (Table 1). This is evidence of the suitable growth conditions for this microbial group resulting from the production technology (low temperature, the presence of salt and nitrite, the vacuum in the tumblers). Results showed a high biodiversity in LAB community, whereas the most frequently recognized species belonged to the genus *Leuconostoc*: *L. carnosum* (35 samples), *L. gelidum* (25 samples), *L. mesenteroides* (10 samples), and *L. inhae* (4 samples). The second largest group of the detected isolates was represented by species of the genus *Latilactobacillus*: *L. sakei* (23 samples, often isolated along with the species *L. gelidum*), *L. curvatus* (6 samples), and *L. fuchuensis* (4 samples). The species *Dellaglioa algida* (formerly *Lactobacillus algidus*) was isolated from 6 samples from one batch of tumbled meat. Among the species only occasionally detected were *Lactococcus lactis* (1 sample) and species of the genus *Lentilactobacillus*, specifically *L. kefiri* (2 samples, formerly *Lactobacillus kefiri*) and *L. otakiensis* (1 sample, formerly *Lactobacillus otakiensis*). *Pancilactobacillus oligofermentans* (formerly *Lactobacillus oligofermentans*) was found in one sample.

### 3.3. Cooked Hams

The heat treatment destroyed the majority of the bacteria present in cooked hams, as the bacterial counts were beneath the limit of detection with a few exceptions. The action of a temperature of 70 °C is sufficient to kill most vegetative bacteria, though sporogenic bacteria are capable of surviving. This is evidenced by the unique finding of 2 samples from 30 analysed hams in which values for the TVC of around 1.40 log CFU/g and 2.99 log CFU/g were recorded (Table 1). Since the presence of neither LAB nor *Enterobacteriaceae* was demonstrated in these two samples, these were most probably sporogenic bacteria. The conditions that exist in cooked hams (practically anaerobic environment, salt content between 1.8–2.2%, presence of nitrite, low temperature) do not support the growth of sporogenic bacteria [16].

Valuable results were obtained from cooked hams (*n* = 12) that were subjected to a “stability test” at 15 °C (Appendix A). Although no bacteria were demonstrated in samples of these hams before the stability test, subsequent testing showed 11 samples with a TVC > 1 log CFU/g, 6 samples with evidence of bacteria of the family *Enterobacteriaceae* > 1 log CFU/g and 11 samples with evidence of LAB > 1.7 log CFU/g (Table 1). Only 1 of the total number of 12 samples was negative for all indicator groups of bacteria. Although samples were taken for bacteriological analysis in a sterile manner from the depth of the cooked hams where secondary contamination was precluded, the presence of bacteria was demonstrated in all samples except one. The results attained point to the fact that the heat treatment of products (core temperature of 70 °C/10 min.) causes sublethal damage in addition to killing bacteria. As stated by Wu [17], damaged cells are potentially just as important as undamaged bacteria, since they can be resuscitated and then capable of normal growth. Therefore, the cold-chain maintenance and avoidance of temperature fluctuations during storage of final products is of great importance in this matter.

Dušková et al. [10] performed the microbiological analysis of individual technological operations during the industrial production of cooked hams focusing on LAB. Their experiment took in the input raw material at the initial stage of production, i.e., at the slaughterhouse, and also focused on critical points in production environment, i.e., the cutting room, the injection and tumbling environment, and final slicing and packing. A reduction to LAB from log 4–5 CFU/g of tumbled meat to a value of practically zero occurred in the cited study during the course of heat treatment. The source of LAB was clearly the input raw material [10]. The data obtained on the presence of LAB in meat following the tumbling process is in agreement with the results attained by Vasilopoulos et al. [8] who found a population of LAB of log 3.22 ± 1.08 CFU/g during the production of cooked hams in small-scale artisanal production in Belgium at the same stage of the production process. The LAB population fell beneath the limit of detection following heat treatment. Following subsequent storage at 7 °C, however, the surviving cells multiplicated to a level of around log 7 CFU/g after 4 w [8]. The growth of LAB during the storage of cooked hams has also been demonstrated by Tamkutė et al. [18]. Kameník et al. [9] found values for the TVC exceeding log 5 CFU/g in cooked hams in barrier casings before slicing. The majority of the bacterial diversity in the cooked hams analysed by Zagdoun et al. [19] was represented by approximately 14 different species in various combinations which taken together represented, on average, 98 % of the total relative bacterial count. The most numerous species in cooked hams were also present in the raw meat after tumbling: *Carnobacterium divergens, Lactobacillus sakei* (now *Latilactobacillus sakei*)*, C. maltaromaticum, Leuconostoc carnosum, L. gelidum,* and *L. mesenteroides.* Heat treatment eliminated only 50% of species present in tumbled meat—generally those that were subdominant [19].

### 3.4. Psychrotrophic or Psychrophilic LAB in Samples

One of the ways in which LAB differ is their ability to grow at refrigeration temperatures. However, there may be a distinction between cold-acclimatized mesophiles and strictly psychrophilic species based on the temperature range they proliferate. *L. sakei, L. curvatus, L. carnosum, L. mesenteroides, Carnobacterium* spp., and *Weissella* spp., for example, belong to the first group, i.e., mesophilic LAB adapted to low temperatures (psychrotrophic). Conversely, *L. gelidum, Dellaglioa algida,* and *L. fuchuensis*, belong to the second category of strictly psychrophilic LAB incapable of growth at 30 °C [20]. Representatives of both groups of LAB were demonstrated in the present study, particularly in input raw material samples. *D. algida* was isolated from the majority of samples of tumbled meat taken from batch no. 4 after incubation at 6.5 °C, but from none of the samples from the same batch cultivated at 15 °C. In the case of *L. gelidum* and *L. fuchuensis*, however, both species were isolated from samples incubated at both 6.5 and 15 °C.

The species demonstrated most frequently in samples of cooked hams was *L. carnosum* (5 positive samples). The authors Vasilopoulos et al. [8] also identified the species *L. carnosum* in almost a half of isolates from samples of cooked hams. Pothakos et al. [21] found identical species in cooked meat products as well. Dušková et al. [10] found the species *L. carnosum, L. mesenteroides,* and *L. gelidum* most frequently in samples of cooked hams following heat treatment. *L. mesenteroides* was demonstrated in three samples of cooked hams in this study and is considered the most rapidly growing species of LAB in cooked hams [6]. Bacteria of the genus *Leuconostoc* are also commonly demonstrated in other cooked meat products [22].

The presence of LAB of the genus *Carnobacterium* could not be demonstrated in this study, neither in the study of Dušková et al. [10], in spite of the fact that these LAB are described as isolates from all whole-muscle meat products [7,8,23,24]. The absence of *Carnobacterium* bacteria during analyses of meat products may be due to exposure to lower temperatures (<12 °C) that positively select leuconostocs and *L. sakei*, which are more psychrotrophic compared to *Carnobacterium* spp. [25]. However, another contributing factor may be the choice of cultivation medium. In this study, the authors used standard MRS agar for the isolation of LAB, while Geeraerts et al. [26] used modified MRS agar (mMRS), in which acetate was missing as a component and the pH value was adjusted to 8.6, for the purpose of better detection of bacteria of the genus *Carnobacterium*.

Pothakos et al. [11] pointed to the fact that a cultivation temperature of 30 °C does not provide an entirely objective picture of the spoilage microbiota present, when evaluating products stored under refrigerated conditions. Dušková et al. [10] found that the use of an incubation temperature of 15 °C resulted in greater LAB detection during analyses of meat and cooked hams than a temperature of 30 °C. An incubation temperature of 15 °C was chosen in the study presented here for this reason. A temperature of 6.5 °C is used for the determination of psychrotrophic bacteria [27]. In the presented study, the temperature of 15 °C provided higher capture of bacteria (*p* < 0.05) except for the determination of the TVC in meat (*p* = 0.06). However, the results were available earlier (after 3 and 6 d) in the case of the use of an incubation temperature of 15 °C than at 6.5 °C (6 and 12 d). It is appropriate to choose an incubation temperature of 15 °C for the detection of indicator bacteria that cause spoilage for this type of foodstuff. Nevertheless, it is also necessary to consider the fact that this temperature need not be suitable for the growth of strictly psychrophilic bacteria, including certain LAB.

## 4. Conclusions

During the production of cooked hams, it is essential to count on an increase in the bacterial contamination of the raw material even if the load on the fresh raw meat is relatively low. A marked increase in the LAB population in the meat occurs during tumbling. Heat treatment causes the destruction of the majority of the microbiota present. It may, however, also induce the merely sublethal damage of vegetative bacteria, particularly certain LAB. A stability test at a temperature of 15 °C can be used for their determination during bacteriological investigation. This method makes it possible to reveal the part of the bacterial population that is capable of shortening the shelf life of the final products under certain conditions. It is appropriate to use a cultivation temperature of 15 °C for the microbiological analysis of meat products for the purpose of detecting the microbiota present, though this does not allow for the detection of certain strictly psychrophilic species. One such example is *Dellaglioa algida*, which was isolated from samples of tumbled meat at a temperature of 6.5 °C, though not at 15 °C. For this reason, the authors recommend using a combination of cultivation temperatures of 6.5 and 15 °C for the comprehensive bacteriological analysis of meat products, including the species differentiation of the LAB present.

Although the primary cultivation of samples of cooked hams did not reveal the presence of LAB, their presence was confirmed in 11 out of 12 samples by a stability test. Bacteria of the genus *Leuconostoc* were the most numerous. Their presence was detected to a limited degree in samples of fresh meat, though their occurrence was greater in meat following tumbling both in quantitative terms and from the perspective of their species representation. The environment of the production plant contributes in this way to the contamination of the production material, and this must be taken into consideration during hygiene measures designed to extend the shelf life of final products.

## Figures and Tables

**Figure 1 microorganisms-10-01106-f001:**
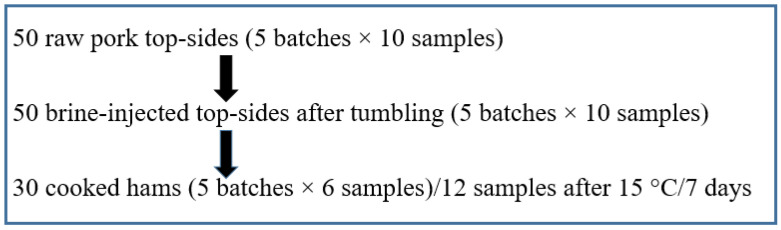
Sampling scheme for bacteriological examination.

**Table 1 microorganisms-10-01106-t001:** Results of determination of the total viable count (TVC), bacteria of the family *Enterobacteriaceae* and lactic acid bacteria (LAB) during incubation at 6.5 and 15 °C in samples of meat, tumbled meat, cooked hams, and samples of cooked hams after a stability test (15 °C/7 d). Results given in log CFU/g and as average, minimum and maximum values.

Analysed Samples	TVC			Enterobacteriaceae			LAB		
	6.5 °C	15 °C	*p*-Value	6.5 °C	15 °C	*p*-Value	6.5 °C	15 °C	*p*-Value
meat (*n* = 50)	3.18 ^a^	3.35 ^a^	0.06	0.51 ^a^	1.07 ^a^	<0.001	<1.70	<1.70	-
	1.00; 4.71	1.00; 4.61		<1.00; 2.04	<1.00; 3.34		<1.70; 3.08 ^III^	<1.70; 2.62 ^IV^	
tumbled meat (*n* = 50)	4.11 ^b^	4.34 ^b^	<0.001	1.00 ^b^	2.14 ^b^	<0.001	2.60 ^V^	3.17 ^VI^	<0.001
	3.52; 4.73	3.59; 5.04		<1.00; 3.23	<2.00; 4.60		<1.70; 3.82	2.18; 3.94	
cooked ham (*n* = 30)	<1.00 ^I^	<1.00 ^II^	-	<1.00	<1.00	-	<1.70	<1.70	-
ham after stability test (*n* = 12)	-	3.64	-	-	1.31	-	-	3.18 ^VII^	-
		<2.00;4.96			<1.00; 3.08			<1.70; 4.76	

^a,b^ different indices in the same column denote statistically significant differences *p* < 0.05; ^I^ Two samples with the finding 1.40 log and 2.99 log CFU/g; ^II^ Two samples with the finding 1.48 log and 2.91 log CFU/g; ^III^ Latilactobacillus sakei, Leuconostoc carnosum; ^IV^ Latilactobacillus curvatus, Latilactobacillus sakei, Leuconostoc gelidum, Enterococcus gilvus; ^V^ Latilactobacillus sakei, Latilactobacillus curvatus, Latilactobacillus fuchuensis, Leuconostoc gelidum, Leuconostoc carnosum, Leuconostoc inhae, Dellaglioa algida; ^VI^ Latilactobacillus curvatus, Latilactobacillus sakei, Latilactobacillus fuchuensis, Leuconostoc carnosum, Leuconostoc gelidum, Leuconostoc mesenteroides, Leuconostoc inhae, Lentilactobacillus kefiri, Lentilactobacillus otakiensis, Paucilactobacillus oligofermentans, Lactococcus lactis; and ^VII^ Latilactobacillus sakei.

## Data Availability

Not applicable.

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
