# Peer review of "The Pork Meat or the Environment of the Production Facility? The Effect of Individual Technological Steps on the Bacterial Contamination in Cooked Hams"

_microorganisms, 2022, doi:10.3390/microorganisms10061106_

Round 1

Reviewer 1 Report

The manuscript describes the limited characterization of the microbial population using plating and MALDI-TOF during production of ham in a commercial production facility. Total counts,  Enterobacteriaceae and lactic acid bacteria (LAB) were determined in the samples at various stages but nothing unusual or novel was observed and therefore the manuscript provides no useful new knowledge or insight.  Some value might be added to the manuscript if additional microbiological quality data for this production facility were included. The company must monitor the microbial quality of the production line on a regular basis and it would be interesting if this information was merged with the more detailed time-limited 'snapshot' analysis presented here. 

A minor point but the putative sporogenic isolates from the cooked ham should be properly identified.

Author Response

Response to Reviewer #1

We would like to thank you for your comments and suggestions that have improved the quality of our manuscript.

“Comments and Suggestions for Authors

The manuscript describes the limited characterization of the microbial population using plating and MALDI-TOF during production of ham in a commercial production facility. Total counts, Enterobacteriaceae and lactic acid bacteria (LAB) were determined in the samples at various stages but nothing unusual or novel was observed and therefore the manuscript provides no useful new knowledge or insight.”

The reviewer is right that the results of the manuscript do not bring much new information about the presence of bacteria in meat or cooked ham. On the other hand, there is valuable knowledge about culture temperatures in microbiological analysis of samples. In many scientific studies, the authors still used a culture temperature of 30 ° C even in the case of samples of meat products, which, however, are not stored at all at relatively high temperatures. For microbiological analysis of meat products, it is necessary to use lower cultivation temperatures, according to our results ideally 15 ° C. However, for some psychrophilic bacteria, also involved in spoilage, cultivation at an even lower temperature is necessary. To the best of our knowledge, the stabilization test described in our study has not yet been used, in which incubation at 15 °C for several days was used to revitalize sublethal damaged bacteria in cooked ham samples. The results provided a completely unique picture of microbial quality, which remained hidden during normal standard bacteriological examinations.

“Some value might be added to the manuscript if additional microbiological quality data for this production facility were included. The company must monitor the microbial quality of the production line on a regular basis and it would be interesting if this information was merged with the more detailed time-limited 'snapshot' analysis presented here.”

The authors of the presented study consider the selected indicator bacteria to be sufficient for occasional monitoring of the production process. Fresh meat analysis is used to select suitable suppliers. Analysis of tumbled meat will provide an overview of the level of processing hygiene. The results of the analysis of cooked hams after heat treatment will provide a picture of the effectiveness of this technological step.

“A minor point but the putative sporogenic isolates from the cooked ham should be properly identified.”

The conditions that exist in cooked hams (practically anaerobic environment, salt content between 1.8-2.2%, presence of nitrite, low temperature) do not support the growth of sporogenic bacteria. For this type of meat products, their analysis does not make sense.

Reviewer 2 Report

In this study, the authors assessed the effect of incubation temperatures of 6.5 and 15 °C on the bacterial burden of fresh, tumbled, and cooked Ham meat. The authors showed that the total bacteria increased from fresh  to tumbled and reduced at cooed meat. The authors recommended using the stability test at a temperature of 15 °C for determination bacterial load during bacteriological investigation.

I have some comments

1- Why the authors choose temperature at 6.5C and 15 C for the study?

2- What are Enterobactericease species detected?

3- Can these procedure and recommendation applied to edible parts of pigs?

4- During the authors detect fungi contamination in these samples?

Author Response

Response to Reviewer #2

We would like to thank you for your comments and suggestions that have improved the quality of our manuscript.

“Comments and Suggestions for Authors

In this study, the authors assessed the effect of incubation temperatures of 6.5 and 15 °C on the bacterial burden of fresh, tumbled, and cooked Ham meat. The authors showed that the total bacteria increased from fresh to tumbled and reduced at cooed meat. The authors recommended using the stability test at a temperature of 15 °C for determination bacterial load during bacteriological investigation.”

“I have some comments

1- Why the authors choose temperature at 6.5C and 15 C for the study?”

Already in our previous studies, the use of a temperature of 15 °C has proved successful in comparison with the usually used cultivation temperature of 30 °C. However, for meat products stored at 2-8 ° C, the use of 30 ° C does not provide suitable conditions. The temperature of 6.5 °C is commonly used to detect psychrotrophic bacteria in food.

 “2- What are Enterobactericease species detected?”

 Enterobacteriaceae represent an important group of indicator bacteria with regard to the hygiene of the production process. Although they play a role in the spoilage of fresh meat (especially at higher pH values), they do not apply to the spoilage of meat products, as they do not survive cooking temperature when properly treated (heat temperature 70 °C).

“3- Can these procedure and recommendation applied to edible parts of pigs?”

They can be applied when using pork for the production of other whole-muscle or comminuted meat products.

“4- During the authors detect fungi contamination in these samples?”

Analyzes were not focused on molds and yeasts. Neither molds nor yeasts grew on any agar during anaerobic cultivation. Mold growth in this type of sample is unlikely. Yeast could be detected in targeted cultivation.

Round 2

Reviewer 1 Report

The authors have not addressed the issue of this study being a 'snapshot' with no contextual data from the facility where the trials were performed.

With respect to the comment 'To the best of our knowledge, the stabilization test described in our study has not yet been used, in which incubation at 15 °C for several days was used to revitalize sublethal damaged bacteria in cooked ham samples', can the authors clarify how this work significantly differs from their use of 15 °C in their previous publication in 2016? 

The identification of the putative sporogenic bacteria in the 2 samples post heating is necessary to confirm that the heat step did eliminate all non-sporeformers particularly as the paper focuses on microbes recovered from heat treated samples.  

Author Response

We would like to thank you for your comments and suggestions that have improved the quality of our manuscript.

“Comments and Suggestions for Authors

The authors have not addressed the issue of this study being a 'snapshot' with no contextual data from the facility where the trials were performed.

With respect to the comment 'To the best of our knowledge, the stabilization test described in our study has not yet been used, in which incubation at 15 °C for several days was used to revitalize sublethal damaged bacteria in cooked ham samples', can the authors clarify how this work significantly differs from their use of 15 °C in their previous publication in 2016?”

Stabilization test (= the samples were vacuum packed and incubated at 15 °C for 7 days before microbial analysis), was used only in this manuscript.

In our previous publication in 2016, we used a temperature of 15 °C to incubate Petri dishes with MRS agar for comparison with a culture temperature of 30 °C. In the present publication, we have already used a culture temperature of 15 °C (and not yet 30 °C) and 6.5 °C for psychrotrophic bacteria.

“The identification of the putative sporogenic bacteria in the 2 samples post heating is necessary to confirm that the heat step did eliminate all non-sporeformers particularly as the paper focuses on microbes recovered from heat treated samples.”

In Chapter 3.3, the following sentences have been added:

“The conditions that exist in cooked hams (practically anaerobic environment, salt content between 1.8-2.2%, presence of nitrite, low temperature) do not support the growth of sporogenic bacteria. For this type of meat products, their analysis does not make sense.”

We did not consider the identification of putative sporogenic bacteria necessary because, according to our previous experience, these bacteria do not continue to grow in the environment of cooked hams after heat treatment.

Reviewer 2 Report

No further comments

Author Response

Thank you for reviewing the manuscript.